

# Pesticide dynamics in three small agricultural creeks in Hesse, Germany

Sarah Betz-Koch[1], Björn Jacobs[2], Jörg Oehlmann[1], Dominik Ratz[1], Christian Reutter[1], Arne Wick[2] and Matthias Oetken[1]

[1] Department Aquatic Ecotoxicology, Johann Wolfgang Goethe Universität Frankfurt am Main, Frankfurt/Main, Germany
[2] German Federal Institute of Hydrology, Bundesanstalt für Gewässerkunde (BfG), Koblenz, Germany

## ABSTRACT

**Background**. Due to their high biodiversity, small water bodies play an important role for freshwater ecosystems. Nonetheless, systematic pesticide monitoring in small creeks with a catchment $<30 \ km^2$ is rarely conducted.

**Methods**. In this study, event-driven water samples were taken from May until November 2017 and March until July 2018 after 20 rain events at three sampling sites with catchment areas of $<27 \ km^2$ in the Wetterau, a region with intensive agriculture in Southern Hesse, Germany. Additionally, enriched extracts of the native water samples from the campaign in 2018 were used for the Microtox assay to determine baseline toxicity to invertebrates over time and sum of toxic units (STU) were calculated to compare the potential toxicity of the samples.

**Results**. Overall, 37 pesticides and 17 transformation products were found, whereby the herbicide metamitron (79 µg/L) showed the highest concentration. Regularly, pesticide concentrations peaked at the time of the highest water level within each sampling event. Within each sampling event maximum pesticide concentration was mostly reached in water samples taken during the first two hours. The sum of the time-weighted mean concentration values of all pesticides was between 2.0 µg/L and 7.2 µg/L, whereby the measured concentrations exceeded their regulatory acceptable concentration (RAC) at 55% of all sampling events for at least one pesticide. The mean $EC_{50}$ values varied between $28.6 \pm 13.1$ to $41.3 \pm 12.1$ REF (relative enrichment factor). The results indicated that several samples caused baseline toxicity, whereby the highest activity was measured at the time of highest water levels and pesticides concentrations, and then steadily decreased in parallel with the water level. Median STUs of invertebrates ranged from $-2.10$ to $-3.91$, of algae/aquatic plants from $-0.79$ to $-1.84$ and of fish from $-2.47$ to $-4.24$. For one of the three sampling sites, a significant linear correlation between baseline toxicity and $STU_{invertebrate}$ was found ($r^2 = 0.48$).

**Conclusion**. The results of the present study suggest that (1) current pesticide monitoring programs underestimate risks posed by the exposure to pesticides for aquatic organisms and (2) pre-authorization regulatory risk assessment schemes are insufficient to protect aquatic environments.

Corresponding author
Sarah Betz-Koch,
s.betz@stud.uni-frankfurt.de

# INTRODUCTION

Water is a vital resource for natural ecosystems as well as human life, and it is therefore described by the Water Framework Directive 2000 (EU-WFD) as ''a heritage which must be protected, defended and treated as such''. About 70% of the earth is covered with water, of which only about 2.5% is freshwater (*Oki & Kanae, 2006*). However, the quality of freshwater is threatened by a variety of anthropogenic impacts such as climate change, pollution from agriculture, industry and households—among others (*Brack et al., 2017*).

However, at least 35% of all European surface waters are in chemically poor conditions and 51% fail to achieve a good ecological status (*EEA, 2018*).

In Germany, the diffuse inputs of nitrate (27.1% of the water bodies do not meet the quality standard) and the increased use of pesticides in agriculture (2.8% of the water bodies do not meet the quality standard) are main reasons for this unsatisfactory chemical status according to the Federal Environment Agency (*Umweltbundesamt, 2017*). Pesticides from agriculture mainly enter water bodies either directly by surface runoff, groundwater inflow and sub-surface drainage systems or indirectly through spray drift (*Reichenberger et al., 2007*; *Liess et al., 1999*). It has been shown that the input of pesticides into water bodies is associated with rainfall events, which are highly variable, and depend also on the time of application, resulting in complex exposure dynamics (*Schulz, 2004*). Especially during rainfall events the concentration of pesticides can increase by a factor of 10 to 100 within hours (*Leu et al., 2004*; *Doppler et al., 2012*; *Petersen et al., 2012*; *Xing et al., 2013*; *Rabiet et al., 2015*; *Lefrancq et al., 2017*). Moreover, small creeks are exposed to high pesticide concentrations due to their immediate vicinity to agricultural farmland and poor dilution potential (*Szöcs et al., 2017*). In addition to pesticides from agriculture, it has been shown that especially in areas with mixed land use (agriculture and urban areas), pesticide inputs from urban areas into surface waters must also be considered during rain events. Pesticides from urban areas mostly enter surface waters *via* point sources, namely wastewater treatment plants, stormwater drains or combined sewer overflows (*Wittmer et al., 2010*). In particular, the dynamics of pesticide exposure, especially the concentration courses after or during rain events and the seasonal variations, in small creeks is rarely considered in national monitoring programs, even though they make up most of the length of a river system with 80% according to *Lorenz et al. (2017)*. National monitoring programs like the European Water Framework Directive (WFD) or the US Clean Water Act are carried out with monthly or weekly grab sampling (*European Union, 2000*; *U.S. Environmental Protection Agency, 1972*), which are conducted completely independent of (heavy) rain events, resulting in underestimated peak exposures (*Bundschuh, Goedkoop & Kreuger, 2014*; *Botta et al., 2012*). Recent studies have shown that peak concentrations in streams can only be detected by water sampling in conjunction with rainfall events (*Vormeier et al., 2023*; *Halbach et al., 2021*; *la Cecilia et al., 2021*; *Lefrancq et al., 2017*).

In 2009, the Directive of the European Parliament and of the Council established a framework for community action to achieve the sustainable use of pesticides (*European Union, 2009*). This framework was incorporated in German national law by the amended Plan Protection Act. The National Action Plan on Sustainable Use of Plant Protection

Products (NAP) includes the aim to consider small creeks for the assessment of pesticide exposure by means of representative monitoring (*Brinke, Bänsch-Baltruschat & Keller, 2017*). An additional aim of the German NAP is that 99% of event-driven samples should not exceed the regulatory acceptable concentration (RAC) values until 2023 (*Federal Ministry of Food and Agriculture of Germany (BMEL), 2013*). In the environmental risk assessment as part of the pesticide authorization regulatory acceptable concentration(RAC) values are used to estimate the potential risks. RACs of pesticides are derived for surface waters, based on available effect data (*EFSA Panel on Plant Protection Products and their Residues (PPR), 2013*). The RAC values can vary between EU member states and can be adapted based on new effect data. In order to rule out a threat to biocenoses, RAC exceedances should not occur in surface waters (*EFSA Panel on Plant Protection Products and their Residues (PPR), 2013*).

Small water bodies play an important role in freshwater ecosystems due to their higher biodiversity compared to larger systems (*Biggs et al., 2014*). Several studies have shown that pesticides can have various effects on aquatic species (*Liess & Von Der Ohe, 2005*; *Beuter et al., 2019*). *Beketov et al. (2013)* determined that pesticides reduce species and family richness of creek invertebrates in Europe (Germany, France) and Australia (Southern Victoria), with losses of taxa up to 42%.

Aquatic ecosystems are normally exposed to a mixture of pesticides, which may result in a higher toxicity due to additive interactions or combined action (*Abdo et al., 2015*). To assess the potential toxicity of a pesticide mixture to aquatic species, toxic units (TUs) can be used. A TU is defined as the logarithmically ratio between a concentration of a pesticide and its respective toxicity ($LC_{50}$) for a species (*Liess & Von Der Ohe, 2005*). For pesticide mixtures, the sum of TUs can be used to assess the potential effect to aquatic species. In addition to STUs an *in-vitro* bioassay (Microtox assay) can be used to assess the toxicity of the enriched water samples. Many studies have shown that this bioluminescence inhibition test as a common method in ecotoxicology is suited to investigate baseline toxicity of environmental samples (*Escher et al., 2008*; *Tang et al., 2013*; *Voelker et al., 2017*; *Parvez, Venkataraman & Murkherji, 2006*).

In 2017 and 2018, a joint investigation was conducted with the aim to monitor pesticide exposures in small creeks (summery outflow <1 $m^3$/s) in a region with intensive agriculture by means of event-driven samplers, including the assessment of the toxicity of these water samples.

Water samples were taken from May until November 2017 and March until July 2018 after rainfall events by an automatically event-driven sampler in several small creeks in the Wetterau region in Southern Hesse, Germany. The samples were analyzed for 37 pesticides and 17 transformation products. To investigate the toxicity of the samples, an *in-vitro* test targeting baseline toxicity (Microtox assay) was applied. In the project, the following hypotheses were tested: (1) Rainfall events lead to high peak concentrations of pesticide in small creeks located in agricultural land, exceeding RAC values for different pesticides; and (2) the pesticide exposure causes baseline toxicity.

## MATERIALS & METHODS

### Sampling location

The Nidda is situated in Southern Hesse and has a length of 89.7 km. The source of the Nidda is located at an altitude of 720 m a.s.l. near Schotten in the area of the Hohen Vogelsberg, and it flows into the Main near to Frankfurt-Höchst. The catchment covers an area of 1,943 km² and it is mainly characterized by intensive agriculture (*HLNUG, 2021*). In detail, the usage of the catchment area is divided into agriculture (∼53%), forests (∼32%), settlements (∼9%), industries (∼2%), green spaces (∼2%), culture and services (∼1%), traffic (<1%), water bodies (<1%) and other usages (<1%) (*Regierungspräsidium Darmstadt, 2015*; *Schulz & Bischoff, 2008*). The Nidda is a highly anthropogenically influenced river with 62 wastewater treatment plants as well as pesticide inputs through agriculture (*Bach, Röpke & Frede, 2005*). With a length of 44.5 km the Horloff is one of the main tributaries of the Nidda. The catchment area of the Horloff is 279 km² (*HLNUG, 2021*). The ecological status of the Horloff is rated from moderate to poor from the upper to the lower river course due to anthropogenic influences like agriculture, settlements and traffic (*HLNUG, 2021*). Therefore, in most flow sections the ecological status did not comply with the requirements of the EU-WFD.

Within the catchment area of the Horloff, three sampling sites were chosen to determine the effects of episodic pesticide inputs after heavy rainfall events. All three creeks are tributaries of the Horloff with a catchment area of <27 km² each (*HMUKLV, 2021*). In addition, the sampling sites were chosen based on the characteristics of proximity to agricultural areas, site accessibility, stable water levels and rising water levels after heavy rainfalls.

### Sampling sites

For the pesticide monitoring three sampling sites were chosen (Fig. 1). The Langder Flutgraben (also kown as Bach- and Biebergraben) is a left-side tributary of the Horloff. The river is a small siliceous low mountain creek rich in coarse material (*HMUKLV, 2021*). The Waschbach is a right-side tributary of the Horloff and is also characterized as a small siliceous low mountain creek rich in coarse material (*HMUKLV, 2021*). The Weidgraben is a silicate low mountain creek rich in fine to coarse material. It should be noted that the creeks Langder Flutgraben and Waschbach are impacted by combined sewer overflows (CSOs) (*HLNUG, 2021*). The location of each event-driven sampler is described in Table S1.

### Sampling and preparation of the native water samples

Water samples were taken by event-driven, mobile samplers (type P6L; MAXX Mess- und Probenahmetechnik GmbH, Rangendingen, Germany). The event-controlled mobile sampler has no cooling function, which is why degradation of the water samples within 12 h cannot be ruled out. However, most target analytes tend to be persistent or degrade slowly. To realize an automatic event-driven sampling after the onset of a rain/discharge event, the samplers were equipped with floating switch and activated after exceedance of

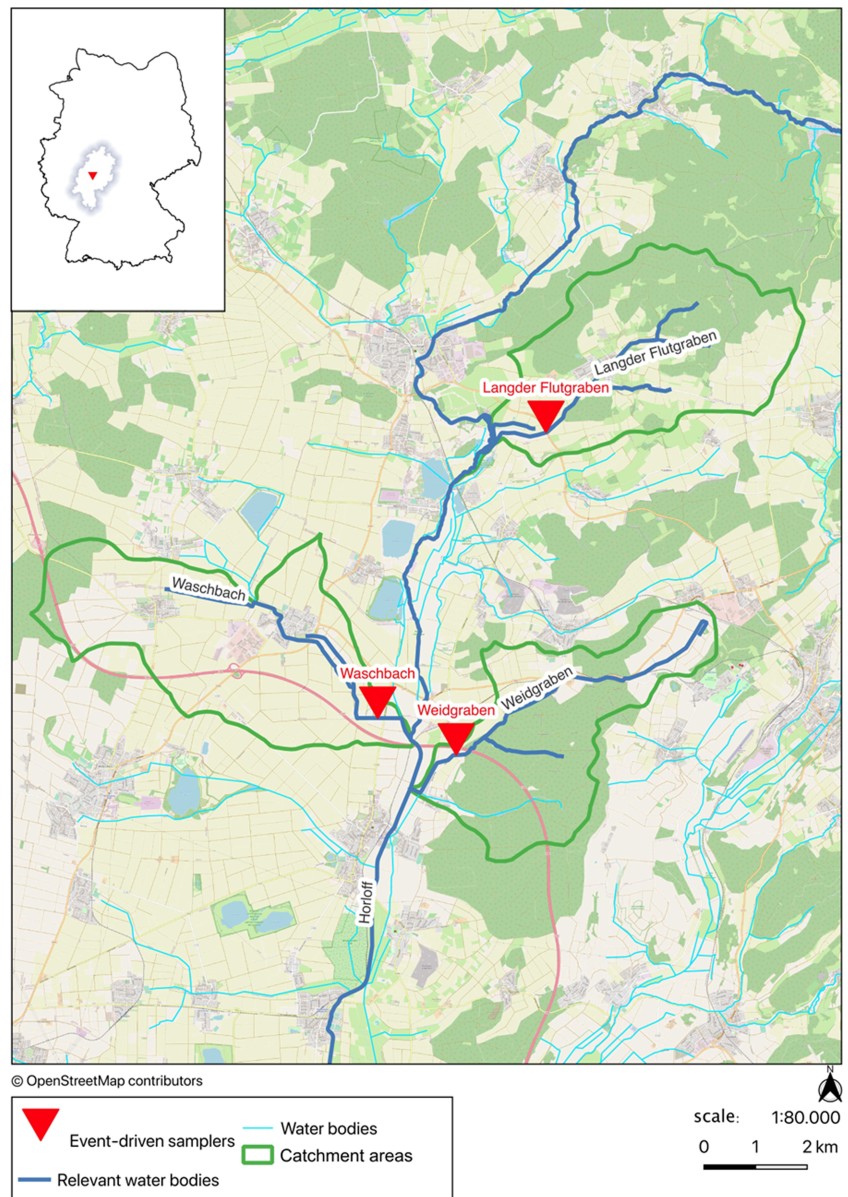

**Figure 1 Overview map of the sampling sites.** This figure was created with data from OpenStreetMap (http://openstreetmap.org) and can be used and modified under CC-by-SA-License (http://creativecommons.org).

a certain water level. The water level was continuously recorded (every 10 min) at each sampling site with a data logger for water level, pressure and temperature (HOBO U20L).

The P6L comprises a housing of 24 glass vessels with a capacity of 300 mL each. For maximal time-resolution, 24 samples were taken over a period of 24 h after activation. The sampling program is summarized in Table S2. The sampling program was divided in three cycles. Since changes in pesticide compositions were assumed to be higher at the beginning, the sampling interval of each sample during the first 3 h (cycle 1) was set to 20

min (9 samples), followed by a time period of 9 h (cycle 2) with a sampling interval of 60 min (9 samples) and finally a time period of 12 h (cycle 3) with a sampling interval of 2 h (6 samples). Each sample comprised of four subsamples of 75 mL each taken every 5 min, 15 min and 30 min during cycle 1, 2 and 3, respectively. Hence, the final sampling volume was 300 mL for each sample. After activation of the sample a text message was generated and samples were picked-up not later than 12 h after the sampling period. In order to achieve sufficient sample volumes for toxicity testing, the water of three consecutively taken samples was poured into a one 1 L amber glass bottle as a composite sample (CS).

For chemical analysis an aliquot of 10 mL was filtered with a 0.45 μm syringe filter (Whatman SPARTAN, regenerated cellulose; GE Healthcare Life Sciences) and transferred into a micro tube on-site. Samples were transported cooled to the lab and stored frozen ($-20$ °C) until analysis. For toxicity testing the remaining sample was transported also cooled to the lab and filtered with grade 696 qualitative filter paper with a retention size of 1.5 μm (VWR International bvba, Leuven, Belgium) within 24 h. 500 mL of each CS was used for *in-vitro* testing with Aliivibro fischeri after conducting a solid phase extraction(SPE), using a C18 column (Telos™ SPE Columns; Kinesis Inc., Vernon Hills, Illinois, USA) to enrich each CS by a factor of 1,000. Prior to SPE the water samples were acidified with 3.5 M sulfuric acid to a pH of 2.5 ($\pm$ 0.1). Subsequently, 500 mL of the CSs were drawn onto the columns *via* vacuum, dried under a gentle nitrogen stream and stored at $-25$ °C until further processing. Columns were eluted with five mL methanol and five mL acetone. The solution was collected in sterilized amber glass vials. After adding 500 μL dimethyl sulfoxide (DMSO, CAS 67-68-5), the combined methanol-acetone extract was concentrated to 500 μL final volume under a gentle nitrogen stream. 500 mL of ultrapure water was used as blanks for each sampling date. Afterwards, the extracts were stored at $-25$ °C.

### Sampling times

The pesticide monitoring took place from May 20, 2017 until November 8, 2017 and March 30, 2018 until August 15, 2018 to cover the main application time of the pesticides in spring and fall (*Liess & Von Der Ohe, 2005*). Overall, samples were collected after 20 rain events. The chemical analysis was performed after each sampling campaign within 6 months. Each sampling event and the samples taken, analyzed and extracted per cycle are presented in Table S3. Figure S1 shows the total precipitation during the monitoring period. Due to technical difficulties, some sampling cycles comprised less than eight CSs per event.

### Chemical analysis

In total 37 pesticides and 17 transformation products (TPs) were analyzed (Table S4) by direct injection of 80 μL into a liquid-chromatography (Agilent 1260 infinity series) system coupled to tandem mass spectrometry (Sciex Triple Quad 6500+) (LC-MS/MS). Additionally, the pharmaceutical ibuprofen was analyzed as a marker of raw wastewater in order to assess whether samples were impacted by combined sewer overflow. Chemical separation was performed on an Agilent Zorbax Eclipse Plus C18 column (Narrow Bore RR, 2.1 × 150 mm, 3.5 μm) with a Zorbax Eclipse XDB-C8 Guard Column (2.1

× 12.5 mm, 5 µm) according to *Hermes et al. (2018)*. Ionization was achieved using alternating electrospray positive ionization mode (ESI+) and negative ionization (ESI−). For compensation of matrix effects 26 stable isotopes labeled surrogate standards were used. Details about the LC gradient program (Table S5), ESI source parameters (Table S6) as well as the retention times and compound-specific MS/MS parameters (Table S7) of all measured analytes and surrogates are shown in the supporting information. The chosen analytical method did not allow to include the insecticide group of pyrethroids and the herbicide glyphosate into the pesticide monitoring.

## Toxic units (TUs)

To compare the toxicity of the measured pesticide concentrations, TUs were calculated (*Liess & Von Der Ohe, 2005*) for invertebrates, algae/aquatic plants and fish. For each compound, TU values were based on acute $EC_{50}$ values for the most sensitive taxon, which were obtained from the Pesticide Properties DataBase (PPDB) operated by the University of Hertfordshire (http://sitem.herts.ac.uk/aeru/iupac/index.htm) and calculated by the following equation:

$$TU_i = \log\left(\frac{C_i}{EC_{50i}}\right) \tag{1}$$

where $TU_i$ = toxic unit of the pesticide $i$, $C_i$ = concentration (ng/L) of the pesticide $i$, and $EC_{50i}$ = $EC_{50}$ for the most sensitive taxon to the substance/pesticide $i$ (ng/L).

After rainfall events, a mixture of various compounds was measured and it was assumed that this would lead to additive toxicity. To compare the toxicity of pesticide mixtures, the sum of toxic units (STUs) was calculated according to *Backhaus & Faust (2012)*:

$$STU = \sum_{i=1}^{n} TU_i \tag{2}$$

where STU = sum of toxic unit and TU($i$) = toxic unit of the pesticide $i$.

$STU_{invertebrate}$, $STU_{algae/aquaticplants}$ and $STU_{fish}$ were calculated for each analyzed CS. To assess the toxicity of invertebrates the $EC_{50}$ (48 h–96 h) values for *Daphnia magna* or *Chironomus* sp. were used. For the assessment of toxicity to aquatic plants the $EC_{50}$ of the acute 7-day test with *Lemna gibba* and for freshwater algae the $EC_{50}$ of the 72 h growth inhibition test of mostly *Scenedemus subspicatus* and *Pseudokirchneriella subcapitata* was used. The toxicity of fish was assessed by using the $LC_{50}$ (96 h, mostly *Oncorhynchus mykiss*) value (Table S8). Afterwards, the effects measured in the Microtox assay were correlated with the STUs as a proxy for the potential toxicity of each CS.

## *In-vitro* testing—baseline toxicity

To assess the baseline toxicity of the enriched water samples, the bioluminescence inhibition assay (Microtox assay) with the bacterium *Aliivibro fischeri* was used. Only the samples from the spring/summer 2018 campaign were used for this test. The test was performed according to ISO 11348-3 (ISO 11348-3, 2007), modified to a 96-well microtiter as described by *Escher et al. (2008)*. The solvent control (DMSO), the positive control (3,5-dichlorophenol; Sigma-Aldrich Chemie GmbH, Steinheim, Germany), the blank, the negative control
and the extracts were diluted (1:2) in a buffer solution. A total of 50 µL of the bacteria solution were stored in the dark for five minutes. Afterwards, the starting luminescence was measured by a microplate reader (Spark 10M; Tecan, Crailsheim, Germany). Subsequently, 100 µL of the sample were transferred to 50 µL of the bacteria suspension. To determine the inhibition, the luminescence was measured again after 30 min of incubation.

## Data analysis
### Chemical analysis
For each analyzed substance, information was added regarding the number of quantifications, limit of quantification (LOQ), RAC based on the UBA-RAC list (*Umweltbundesamt, 2020*), exceedances of RACs and duration of RAC exceedances. In total, RAC values were available for 36 substances corresponding to 67.9% of the pesticides analyzed. Additionally, the time-weighted mean concentration (TWM) of the CSs for each creek were calculated by using the following equation:

$$TWM_i = \sum_{i=1}^{n} \left( \frac{C_i}{24 * 60} \right) \cdot (t_{CS}) \tag{3}$$

where $TWM_i$ = time-weighted mean concentration of the pesticide $i$, $C_i$ = concentration (ng/L) of the pesticide $i$, $t_{CS}$ = time period (min) of each composite sample.

### Baseline toxicity
Effect concentrations (ECs) corresponding to the relative enrichment factor (REF) were derived from a non-linear regression using a four parameter logistic function. The results of the Microtox assay are expressed as $EC_{50}$, referring to the REF needed to inhibit the bioluminescence by 50%. By combining the measured activities of three independent test runs with eight replicates each, the determination of REF $EC_{50}$ values including a 95% confidence interval (CI) was performed with GraphPad Prism© 5.03. An $EC_{50}$ REF of 52.9 was defined as a threshold for samples inhibiting less than 20% luminescence and consequently not being toxic.

## Statistical analysis
Statistical analysis was conducted with GraphPad Prism© 5.03. First, the data were checked for normal distribution (Shapiro–Wilk Test). With a given normal distribution, the data were considered with a parametric Levene's test for variance homogeneity. When data showed variance homogeneity, an ANOVA followed by a post-hoc test (Tukey-Kramer-HSD) was performed. If the data did not show a normal distribution, a Kruskal-Wallis test followed by a post-hoc test (Dunn's Multiple Comparison Test) was conducted, whereby $p < 0.05$ was considered as significant. To determine a relationship between two variables a Pearson correlation was conducted.

# RESULTS
## Chemical analysis
Overall, CS of 20 rainfall events from three sampling sites were analyzed. Table 1 shows the number of rainfall events for each sampling site and their seasonal distribution. We
**Table 1** Number of sampling campaigns for each sampling site and their seasonal distribution.

| Creek | Number of sampling campaigns | | |
|---|---|---|---|
| | Spring 2017 | Fall 2017 | Spring 2018 |
| Langder Flutgraben | – | – | 5 |
| Waschbach | 3 | 3 | 4 |
| Weidgraben | – | 3 | 2 |

found 37 pesticides (23 herbicides, nine fungicides, five insecticides) and 17 transformation products (TPs). Hence, except for aclonifen, acetamiprid, prochloraz, and DCPU, each of the measured pesticides and TPs was detected at least once during the monitoring campaign in at least one creek. Figure S2 shows the total number of different pesticides detected per group for each creek.

Table 2 provides an overview of all substances with their time-weighted mean and maximal concentration measured as well as their frequency of detection ($FOD_{LOQ}$) during heavy rainfall events. Herbicides were detected most frequently and at the highest concentration at all sampling locations. With 78 µg/L, metamitron was detected with the highest maximum concentration (Waschbach, May 31, 2017), followed by (S)-metolachlor (15 µg/L; Langder Flutgraben, 31 May, 2018), prosulfocarb (13 µg/L; Langder Flutgraben, 31 May, 2018) and terbuthylazine (8.7 µg/L; Langder Flutgraben, 31 May, 2018). The sum of TMWs of all detected pesticides during all sampling events was between 2 µg/L (Weidgraben) and 7.2 µg/L (Langder Flutgraben).

From Table 2, a creek-specific occurrence of certain pesticides can be recognized. Bifenox-free acid was only detected above the LOQ at Weidgraben, while concentrations of difenoconazole, fenpropimorph, fluazifop, imidacloprid, irgarol, mecoprop, terbutryn and triadimenol above their LOQs were only detected at the Waschbach and Langder Flutgraben creeks. In addition, the pesticides chlortoluron, and napropamide as well as the TP dimethenamide-OA were detected above their LOQs at the Waschbach and Weidgraben creeks but not at Langder Flutgraben. Furthermore, concentrations of clothianidin and chloridazone above their LOQs were exclusively detected at the Waschbach creek.

## Seasonal variations

To determine whether seasonal variations exist in detected pesticides, TWM from Weidgraben and Waschbach were compared between the fall 2017 ($TWM_{fall}$) and spring/summer 2018 ($TWM_{spring/summer}$) sampling campaigns (Table 3). Since the Langder Flutgraben was not included in the monitoring program until 2018, there were no data available for fall 2017. The fall 2017 sampling campaign contains three rainfall events (September 18, 2017; October 4, 2017 and November 8, 2017) for both creeks. For Waschbach three rainfall events (May 15, 2018; May 24, 2018 and June 16, 2018) and for Weidgraben two events (June 12, 2018 and August 15, 2018) are included in the spring/summer campaign 2018. As shown in Table 3, the $TWM_{fall}$ of dimethachlor and metazachlor and their respective TPs were generally higher than the $TWM_{spring/summer}$ at Weidgraben and Waschbach. Besides, as shown in Table 3 the $TWM_{fall}$ of dimethenamide, dimethomorph, flurtamone, isoproturon and quinmerac were higher

Table 2 Time-weighted mean (TWM) and maximal concentrations as well as the frequency of detection (FOD$_{LOQ}$) of all detected pesticides during all sampling events.

| Substance | LOQ [ng/L] | Langder Flutgraben (n = 5) [ng/L] | | | Waschbach (n = 10) [ng/L] | | | Weidgraben (n = 5) [ng/L] | | |
|---|---|---|---|---|---|---|---|---|---|---|
| | | TWM[a] | Max[b] | FOD$_{LOQ}$ [%] | TWM[a] | Max[b] | FOD$_{LOQ}$ [%] | TWM[a] | Max[b] | FOD$_{LOQ}$ [%] |
| Bifenox-free acid | 20 | <LOQ | <LOQ | 0 | <LOQ | <LOQ | 0 | <LOQ | 79 | 20 |
| Carbendazim | 1 | 2.6 | 14 | 80 | 8.8 | 44 | 100 | <LOQ | 3.7 | 20 |
| Clomazone | 2 | <LOQ | <LOQ | 0 | 3.6 | 31 | 90 | 3.7 | 17 | 40 |
| Clothianidin | 5 | <LOQ | <LOQ | 0 | 5.4 | 98 | 30 | <LOQ | <LOQ | 0 |
| Chloridazone | 20 | <LOQ | <LOQ | 0 | 53 | 1,800 | 40 | <LOQ | <LOQ | 0 |
| Chlortoluron | 2 | <LOQ | <LOQ | 0 | 2.7 | 10 | 40 | <LOQ | 12 | 20 |
| Difenoconazole | 1 | <LOQ | 5.6 | 60 | 1.4 | 9.2 | 90 | <LOQ | <LOQ | 0 |
| Diflufenican | 1 | 16 | 47 | 100 | 2.9 | 13 | 100 | <LOQ | 2 | 40 |
| Dimethachlor | 2 | <LOQ | <LOQ | 0 | 10 | 140 | 100 | 21 | 230 | 60 |
| Dimethachlor-ESA[c] | 5 | 9.5 | 14 | 100 | 15 | 53 | 100 | 240 | 1,400 | 100 |
| Dimethachlor-OA[d] | 5 | <LOQ | 5.1 | 20 | 6.9 | 46 | 40 | 180 | 1,100 | 80 |
| Dimethenamide | 2 | 9.9 | 65 | 100 | 65 | 740 | 100 | 16 | 160 | 100 |
| Dimethenamide-ESA[c] | 10 | 12 | 35 | 60 | 19 | 54 | 100 | 29 | 120 | 80 |
| Dimethenamide-OA[d] | 20 | <LOQ | <LOQ | 0 | <LOQ | 30.9 | 10 | <LOQ | 79 | 60 |
| Dimethomorph | 2 | 2.2 | 17 | 20 | 9.7 | 103 | 90 | <LOQ | 3.5 | 60 |
| Diuron | 1 | <LOQ | 7.9 | 60 | 4.1 | 20.0 | 90 | <LOQ | 1.2 | 40 |
| DCPMU[e] | 0.5 | <LOQ | 1.1 | 40 | 0.8 | 2.7 | 90 | <LOQ | 0.5 | 20 |
| Epoxiconazole | 1 | 26 | 140 | 100 | 23 | 230 | 100 | 4.8 | 29 | 100 |
| Fenpropimorph | 2 | 4.3 | 87 | 60 | <LOQ | 21 | 30 | <LOQ | <LOQ | 0 |
| Fluazifop | 5 | <LOQ | 12 | 40 | 5.1 | 120 | 80 | <LOQ | <LOQ | 0 |
| Flufenacet | 2 | 14 | 140 | 100 | 64 | 950 | 100 | 7.7 | 43 | 80 |
| Flufenacet-ESA | 5 | 12 | 72 | 100 | 6.3 | 24 | 60 | 8.5 | 50 | 100 |
| Flufenacet-OA | 10 | 17 | 59 | 100 | <LOQ | 30 | 60 | <LOQ | 48 | 60 |
| Flurtamon | 1 | 24 | 96 | 100 | 3.9 | 81 | 90 | <LOQ | 1.0 | 20 |
| Imidacloprid | 5 | <LOQ | 5.0 | 20 | <LOQ | 37 | 30 | <LOQ | <LOQ | 0 |
| Irgarol | 0.5 | <LOQ | <LOQ | 0 | 0.9 | 9.4 | 90 | <LOQ | <LOQ | 0 |
| Isoproturon | 0.5 | 1.5 | 8.4 | 80 | 28 | 560 | 100 | 2.5 | 40 | 60 |
| Mecoprop | 2 | 4.5 | 38 | 60 | 34 | 250 | 90 | <LOQ | <LOQ | 0 |
| Metamitron | 10 | <LOQ | <LOQ | 0 | 2,600 | 78,000 | 80 | 210 | 1,300 | 40 |
| Desamino-Metamitron | 5 | <LOQ | 21 | 80 | 400 | 5,100 | 90 | 22 | 120 | 60 |
| Metazachlor | 1 | 2.8 | 18 | 80 | 340 | 6,800 | 100 | 49 | 330 | 60 |
| Metazachlor-ESA[c] | 20 | 220 | 510 | 100 | 220 | 670 | 100 | 370 | 1,900 | 100 |
| Metazachlor-OA[d] | 20 | 59 | 200 | 100 | 120 | 380 | 100 | 250 | 1,200 | 100 |
| Metolachlor | 1 | 2,400 | 15,000 | 100 | 180 | 3,900 | 100 | 27 | 220 | 100 |
| Metolachlor-ESA[c] | 2 | 180 | 650 | 100 | 45 | 280 | 100 | 86 | 280 | 100 |
| Metolachlor-OA[d] | 5 | 180 | 700 | 100 | 19 | 260 | 100 | 11 | 77 | 80 |
| Napropamide | 10 | <LOQ | <LOQ | 0 | <LOQ | 110 | 30 | <LOQ | 25 | 40 |
| Propiconazole | 2 | 29 | 180 | 100 | 12 | 250 | 100 | <LOQ | 4.6 | 40 |
| Propyzamide | 1 | 1.0 | 2.1 | 100 | 4.2 | 37 | 100 | 1.3 | 13 | 80 |

**Table 2** (*continued*)

| Substance | LOQ [ng/L] | Langder Flutgraben (n = 5) [ng/L] | | | Waschbach (n = 10) [ng/L] | | | Weidgraben (n = 5) [ng/L] | | |
|---|---|---|---|---|---|---|---|---|---|---|
| | | TWM[a] | Max[b] | FOD$_{LOQ}$ [%] | TWM[a] | Max[b] | FOD$_{LOQ}$ [%] | TWM[a] | Max[b] | FOD$_{LOQ}$ [%] |
| Prosulfocarb | 50 | 2,100 | 13,000 | 80 | 190 | 3,900 | 50 | <LOQ | 170 | 20 |
| Prothioconazole-desthio | 5 | 66 | 260 | 80 | 45 | 290 | 70 | 6.2 | 43 | 80 |
| Quinmerac | 5 | <LOQ | 7.7 | 40 | 260 | 3,400 | 100 | 200 | 1,400 | 80 |
| Tebuconazole | 1 | 20 | 110 | 100 | 36 | 210 | 100 | 13 | 100 | 100 |
| Terbutryn | 1 | 3.7 | 15 | 80 | 160 | 2,400 | 100 | <LOQ | <LOQ | 0 |
| Terbuthylazine | 1 | 1,500 | 8,700 | 100 | 41 | 260 | 100 | 21 | 150 | 100 |
| 2-Hydroxy-Terbuthylazine | 1 | 46 | 220 | 100 | 39 | 370 | 100 | 5.7 | 17 | 100 |
| Desethyl-terbuthylazine | 1 | 97 | 470 | 100 | 25 | 150 | 100 | 8.3 | 87 | 100 |
| Thiacloprid | 2 | 4.9 | 26 | 80 | 21 | 700 | 60 | <LOQ | 4.2 | 40 |
| Thiamethoxam | 5 | <LOQ | 5.6 | 20 | 7.4 | 98 | 40 | 15 | 92 | 20 |
| Triadimenol | 20 | <LOQ | 23 | 20 | <LOQ | 59 | 40 | <LOQ | <LOQ | 0 |

**Notes.**
[a]Time-weighted mean value of the concentration of the composite samples
[b]Maximum value of the concentration in each composite sample
[c]ESA, ethanesulfonic acid
[d]OA, oxalic acid
[e]DCPMU, N-(3,4-dichlorophenyl)-N-methylurea)

than the TWM$_{spring/summer}$ at both creeks. The opposite was observed for metamitron, where the TWM$_{spring/summer}$ were generally higher than TWM$_{fall}$ at both creeks (Table 3). Similar observations were made for thiacloprid and thiamethoxam, where the TWM$_{fall}$ of thiacloprid and thiamethoxam were below their limit of quantification, whereas in spring/summer concentrations above its RAC values were observed. Additionally, the TWM$_{spring/summer}$ of prosulfocarb and terbuthylazine were generally higher than the TMW$_{fall}$.

## Concentration courses and RAC exceedances

As exemplified in Fig. 2, the maximum concentration of each pesticide in the streams was mostly reached in the water samples taken during the first 2 h of the water rise, and then decreased steadily in parallel with the decrease in water level. However, at the creeks Langder Flutgraben and Waschbach the concentration courses of metazachlor-ESA and metazachlor-OA achieved their maximum concentration mostly at the end of a sampling cycle or were more or less at the same concentration range during the sampling cycle (Fig. S3).

Overall, RAC exceedances of maximum concentrations were detected at all three sampling sites in 55% of sampling events (11 out of 20). In total, 11 compounds corresponding to 31% of the measured pesticides for which RAC values are available, exceeded their RAC values at least once. Figure 3 shows the number of detected RAC exceedances for each of these 11 pesticides. Most frequently the RAC was exceeded for the insecticide thiacloprid (10 times corresponding to 50% of all sampling events) followed by metolachlor and prosulfocarb (four times corresponding to 20% of all sampling events), terbuthlylazine, thiamethoxam, diflufenican and chlothianidin (three times corresponding to 15% of all sampling events), metamitron (two times corresponding to 10% of all

**Table 3  Time-weighted mean values of selected pesticides during the fall 2017 (TWM$_{fall}$) and spring/summer 2018 (TWM$_{spring/summer}$) sampling campaign.**

| Substance | LOQ [ng/L] | Fall 2017: Waschbach (n = 3) [ng/L] TWM$^a_{fall}$ | Spring/Summer 2018: Waschbach (n = 3) [ng/L] TWM$^a_{spring/summer}$ | Fall 2017: Weidgraben (n = 3) [ng/L] TWM$^a_{fall}$ | Spring/Summer 2018: Weidgraben (n = 2) [ng/L] TWM$^a_{spring/summer}$ |
|---|---|---|---|---|---|
| Dimethachlor | 2 | 20 | 7.3 | 34 | <LOQ |
| Dimethachlor-ESA[b] | 5 | 19 | 16 | 380 | 37 |
| Dimethachlor-OA[c] | 5 | 12 | 7.6 | 300 | 9.2 |
| Dimethenamid | 2 | 53 | 27 | 23 | 8.7 |
| Dimethenamid-ESA[b] | 10 | 21 | 14 | 45 | <LOQ |
| Dimethenamid-OA[c] | 20 | <LOQ | <LOQ | 24 | <LOQ |
| Dimethomorph | 2 | 28 | <LOQ | 2.2 | <LOQ |
| Flurtamone | 1 | 27 | 2.1 | <LOQ | <LOQ |
| Isoproturon | 0,5 | 59 | 3.1 | 12 | <LOQ |
| Metamitron | 10 | 11 | 170 | <LOQ | 530 |
| Desamino-metamitron | 5 | 18 | 64 | <LOQ | 52 |
| Metazachlor | 1 | 58 | 8.1 | 81 | <LOQ |
| Metazachlor-ESA[b] | 20 | 210 | 170 | 530 | 120 |
| Metazachlor-OA[c] | 20 | 120 | 76 | 400 | 36 |
| Prosulfocarb | 50 | <LOQ | <LOQ | <LOQ | 96 |
| Quinmerac | 5 | 61 | 26 | 330 | <LOQ |
| Terbuthylazine | 1 | 9.5 | 43 | 5.6 | 45 |
| Thiacloprid | 2 | 7.8 | 65 | <LOQ | 3.4 |
| Thiamethoxam | 5 | 20 | 14 | <LOQ | 67 |

**Notes.**
[a] Time-weighted mean value of the concentration of the composite samples
[b] ESA, ethanesulfonic acid
[c] OA, oxalic acid

sampling events) and imidacloprid, metazachlor and flufenacet (1 time corresponding to 5% of all sampling events).

Looking at the RAC exceedances at each sampling location, Waschbach showed the highest amount of RAC exceedances. In 70% of all sampling events the RAC value of at least one pesticide was exceeded. The RAC value of at least one pesticide was exceeded in 60% of all sampling events at the Langder Flutgraben, followed by 20% of all sampling events at the Weidgraben.

The most frequent RAC value exceedances were recorded during the sampling campaigns in May and June of 2017 and 2018 (Table S9). The detected concentrations of (S)-metolachlor, prosulfocarb, terbuthylazine and thiacloprid in May were usually above their RAC value despite decreasing concentrations within the complete sampling cycle of 24 h (Fig. 2).

The RAC value of thiacloprid was exceeded the longest with a duration of 24 h per sampling cycle in two rain events in May (May 15, 2018 at Langder Flutgraben and May 24, 2018 at Waschbach) (Table S9). Thereby, the RAC value of thiacloprid was exceeded up to

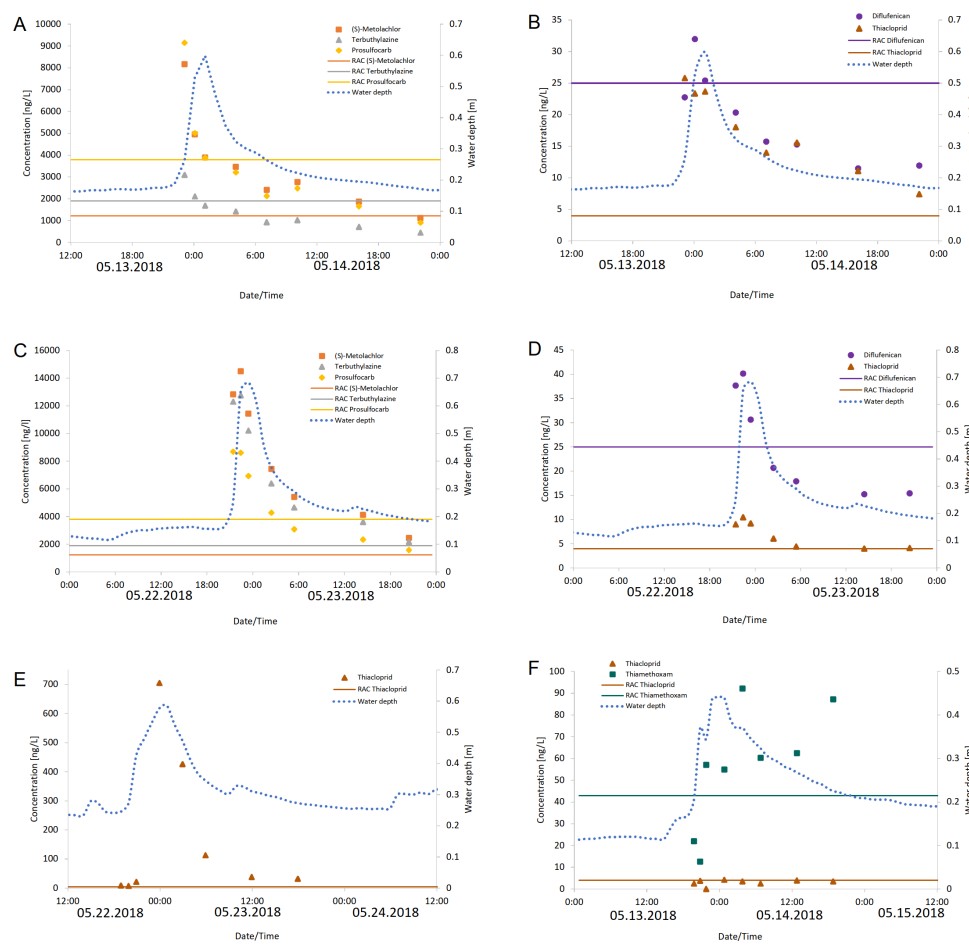

**Figure 2 Measured concentrations of pesticides that exceeded their RAC values.** Measured concentrations of the pesticides diflufenican, (S)-metolachlor, prosulfocarb, terbuthylazine and thiacloprid that exceeded their RAC values at Langder Flutgraben at two sampling events (A, B: 05/13-05/14/2018; C, D: 05/22 –05/23/2018) with the corresponding water depths. Measured concentrations of the pesticide thiacloprid that exceeded their RAC value at Waschgraben (E: 05/22 –05/24/2018) with the corresponding water depth. Measured concentrations of the pesticides thiamethoxam and thiacloprid that exceeded their RAC values at Weidgraben (F: 05/13 –05/15/2018) with the corresponding water depth. The solid horizontal lines provide the RAC value for each pesticide.

a factor of 176, followed by a factor of 11.9 for (S)-metolachlor. Altogether, the duration of RAC exceedances per sampling cycle varied between each pesticide, sampling cycle and sampling time. For example, no RAC exceedance was listed for each of the three sampling cycles conducting in fall 2017 at the Waschbach and Weidgraben creeks. Additionally, no RAC exceedances were recorded for the pesticide thiacloprid in March 2018 at Waschbach but for the sampling in May 2018 RAC exceedances were determined, whereby the duration of RAC exceedances lasted 21 h (May 15, 2018) and 24 h (May 24, 2018) per sampling event, respectively (Table S9).
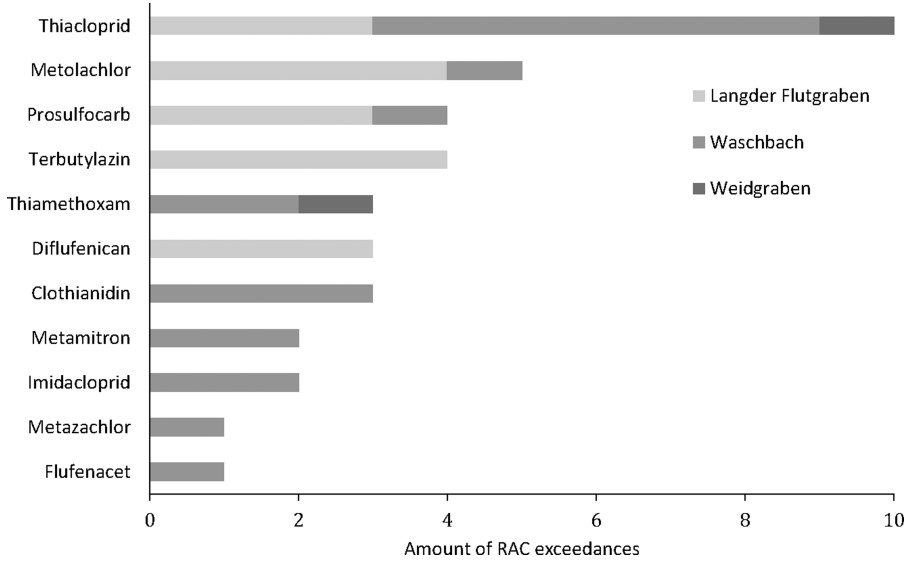

**Figure 3   Number of RAC exceedances during 20 sampling events at three creeks.**

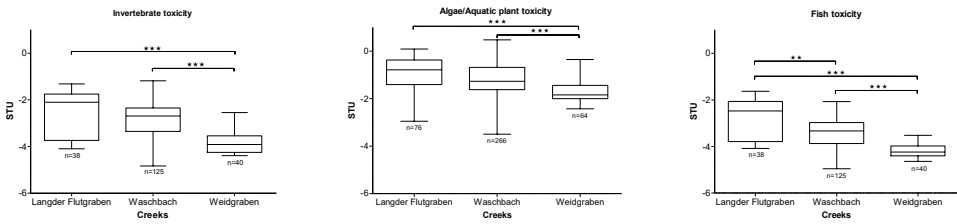

**Figure 4   Sum of toxic units (STUs) for invertebrates, algae/aquatic plants and fish at the three creeks presented as boxplots (median, min to max).** (Kruskal-Wallis Test with Dunn's Multiple Comparison Test, \*\*$p < 0.01$, \*\*\*$p < 0.0001$).

## Toxic units

To illustrate the toxicity of the pesticides in the three creeks, the sums of toxic units (STUs) were calculated for invertebrates, algae/aquatic plants and fish (Fig. 4).

For the Langder Flutgraben creek, the median STU for invertebrates was -−2.10 (25 percentile −3.74; 75 percentile −1.76), for algae/aquatic plants −0.79 (25 percentile −1.41; 75 percentile −0.37) and for fish −2.47 (25 percentile −3.79; 75 percentile −2.06). Thus, the Langder Flutgraben creek is rated as the most toxic. The median STUs of Langder Flutgraben for invertebrates, algae/aquatic plants and fish are also significantly different from the STUs of Waschbach (STU$_{invertebrate}$ −2.69; 25 percentile −3.36; 75 percentile −2.35; STU$_{algae/aquaticplant}$ −1.27; 25 percentile −1.62; 75 percentile −1.44; STU$_{fish}$ −3.34; 25 percentile −3.87; 75 percentile −2.97) and Weidgraben (STU$_{invertebrate}$ −3.91; 25 percentile −4.25; 75 percentile −3.54; STU$_{algae/aquaticplant}$ −1.84; 25 percentile −2.00; 75 percentile −1.44; STU$_{fish}$ −4.24; 25 percentile −4.40; 75 percentile −3.98) (Kruskal-Wallis-Test with Dunn's Multiple Comparison Test, $p < 0.0001$). Additionally, the median STU

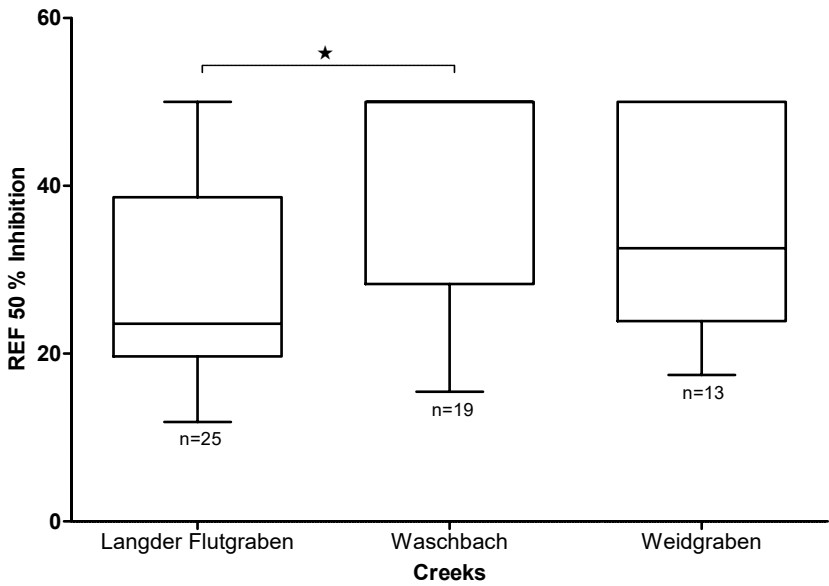

**Figure 5** **Baseline toxicity of the different creeks (median, min to max).** Mean EC50 values are expressed as relative enrichment factors (REF). The EC50 REF of Langder Flutgraben is significantly lower than the EC50 REF of Waschbach (Kruskal-Wallis-Test with Dunn's Multiple Comparison Test, *p = 0.0116).

of Langder Flutgraben for fish is significantly different from the $STU_{fish}$ of Waschbach (Kruskal-Wallis-Test with Dunn's Multiple Comparison Test, $p < 0.01$). The median $STU_{invertebrate}$, $STU_{algae/aquaticplants}$ and $STU_{fish}$ of Waschbach are significantly different from the STUs of Weidgraben (Kruskal-Wallis-Test with Dunn's Multiple Comparison Test, $p < 0.0001$).

### *In-vitro* tests with *Aliivibrio fischeri*

As shown in Fig. 5, the mean $EC_{50}$ value of Langder Flutgraben is $28.6 \pm 13.1$ REF (median 23.6; 25 percentile 19.7; 75 percentile 38.6). Waschbach has a mean $EC_{50}$ value of $41.3 \pm 12.1$ REF(median 50.0; 25 percentile 28.3; 75 percentile 50.0) and Weidgraben of $36.5 \pm 13.1$ REF (median 32.6; 25 percentile 23.9; 75 percentile 50.0). The $EC_{50}$ REF of Langder Flutgraben is significantly lower than the $EC_{50}$ REF of Waschbach (Kruskal-Wallis Test with Dunn's Multiple Comparison Test, $p = 0.0116$).

$EC_{50}$ REF values of Langder Flutgraben correlate with the peak of the water level during heavy rain events. As shown in Fig. 6, the enriched extracts with the highest baseline toxicity, or those requiring the lowest enrichment factor, were those collected at the time of the highest water level. For Waschbach and Weidgraben no correlation between the $EC_{50}$ REF values and the water level peak was found.

By comparing the $EC_{50}$ REF with the STUs calculated for each creek, the strongest correlation was found for Langder Flutgraben and its calculated $STU_{invertebrate}$ ( $r^2 = 0.48$,

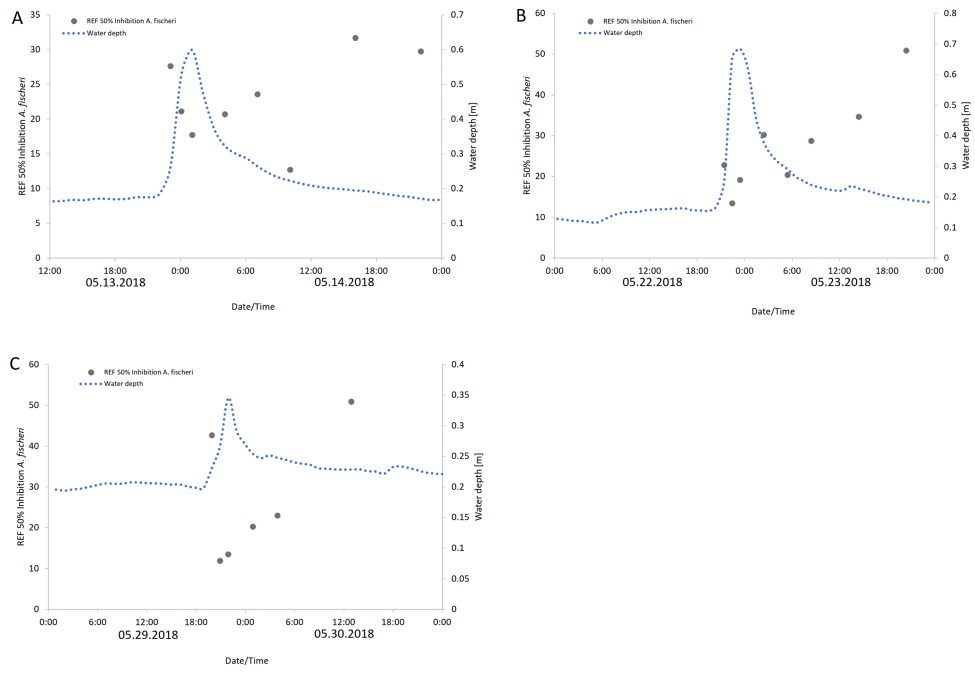

**Figure 6** Baseline toxicity ($EC_{50}$) values are expressed as relative enrichment factors (REF) at Langder Flutgraben at three rain events over a period of time with the corresponding water depths. (A) 05/13 – 05/14/2018; (B) 05/22 –05/23/2018; (C) 05/29 –05/30/2018).

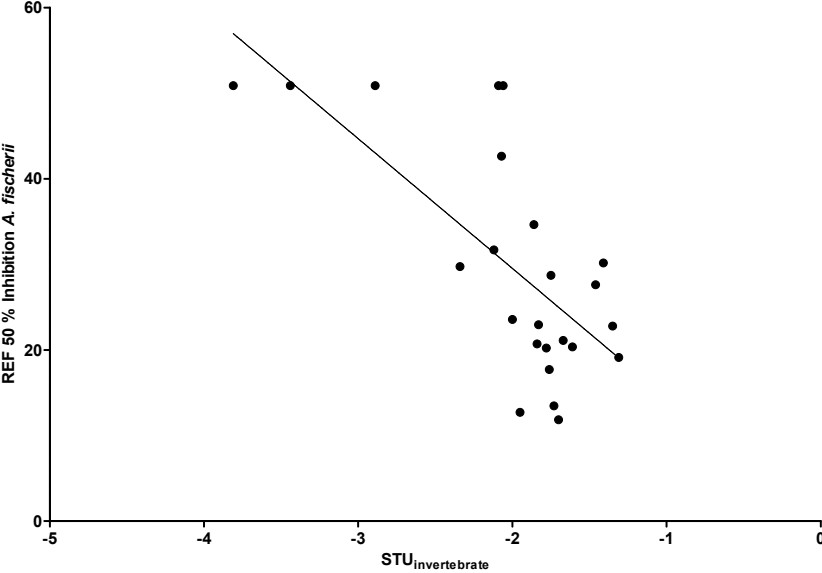

**Figure 7** Linear correlation between $EC_{50}$ REF and STU for the Langder Flutgraben creek ($r^2 = 0.48$, $n = 24$, $p < 0.01$).

Fig. 7). For the other creeks, Pearson product-moment correlation coefficients were not significant.

## DISCUSSION

### Pesticide exposure patterns

Our monitoring program comprised 20 sampling events triggered by water level increase due to rain events at three creeks for 57 substances. *Halbach et al. (2021)* investigated 103 small agricultural creeks for 76 pesticides by taking event-driven water samples and obtained similar results. Moreover, terbuthylazine (0.56 µg/L, 95 percentile, median = 0.0056 µg/L) and (S)-metolachlor showed a high median concentration, and terbuthylazine, flufenacet, prosulfocarb and (S)-metolachlor had the highest frequencies of detection ($FOD_{LOQ}$) of >40% (*Halbach et al., 2021*). Pesticide loss to stream water in a small agricultural catchment in Sweden was studied by *Kreuger (1998)*. In this study, 38 pesticides were detected in water samples over six years (spring 1990 to fall 1996) at daily or weekly intervals during the main application periods. The highest time-weighted mean concentration during a single week was detected for metamitron (24 µg/L) and metazachlor (200 µg/L) (*Kreuger, 1998*). The concentrations of the detected pesticides were mostly in the same ranges as in our study, suggesting similar cropping practices. Furthermore, this indicates that the data from the current study are transferable to regions with similar climate conditions, land use, and management. Overall, the creeks Langder Flutgraben and Waschbach showed higher numbers of pesticides as well as higher concentrations compared to Weidgraben. One reason could be that in contrast to Weidgraben these two creeks are impacted by CSOs (*HLNUG, 2021*). Discharge of CSO in Langder Flutgraben and Waschbach during most of the sampled rain events could be confirmed by the measurement of increased concentrations of the pharmaceutical ibuprofen of up to 1 µg/L. *Wittmer et al. (2010)* confirmed that in catchment areas with mixed land use (agricultural and urban uses), pesticide inputs to surface waters from urban areas play an important role and should not be neglected, especially during rain events. It has been highlighted that diuron and mecoprop—which are used for weed control on roadsides or as protective agents for roofs—showed their highest concentrations during rain events (*Wittmer et al., 2010*; *Gasperi et al., 2008*). This is confirmed by our study, where—for example—the biocides diuron, carbendazim and terbutryn which are not approved for use as plant protection product in Germany were detected at higher concentrations at the creeks affected by a CSO. Mecoprop and the pharmaceutical ibuprofen were even measured below the limit of quantification at Weidgraben, likely as a consequence of the lack of a CSO. Moreover, also pesticides used predominantly or even exclusively as plant protection products in agriculture such as metamitron or metazachlor can be emitted directly by CSO into streams during rain events and not *via* a wastewater treatment plant as is normally the case if a critical discharge in the sewer system is exceeded (*Mueller et al., 2002*) . However, for assessing the contribution of CSO to the measured concentrations of these plant protection products compounds loads emitted by CSO need to be determined which was out of the scope of the current study.

Besides this, *Halbach et al. (2021)* reported that pesticides show a significant correlation with the main crop types grown in the respective catchment area. Additionally, *Spycher et al. (2018)* confirmed that crop types grown were more relevant than the catchment size. In the Waschbach catchment area there are many agricultural areas where beets are grown (*HLNUG, 2021*). The herbicide metamitron is mostly applied to this crop type (*Mueller et al., 2002*) which could explain, for example, the detected high levels of metamitron in the creek after rain events. The high concentrations of (S)-metolachlor, prosulfocarb and terbuthylazine are most likely a result of the widespread cultivation of corn in this area (*Spycher et al., 2018*; *Halbach et al., 2021*; *HLNUG, 2021*).

Additionally, we observed seasonal variations in the occurrence of individual pesticides. This mostly depends on the application time, mass as well as the frequency on adjacent agricultural fields (*Lorenz et al., 2017*). In our study, herbicides and fungicides were detected from fall to spring/summer, whereby insecticides were mostly measured in spring/summer. Similar results were observed in small Swedish lowland agricultural streams, where the detection frequency of about 95% of herbicides and fungicides from March until October was monitored in time-integrated samples while insecticides were mainly found in summer months, corresponding to the respective application period (*Lorenz et al., 2017*). *Vormeier et al. (2023)* also investigated a direct temporal relationship between the application period, the associated rainfall events and the pesticide peaks observed in the catchments, with the relative rate of pesticide application increasing from March to May and then decreasing again until August. In addition, individual properties of each pesticide such as water solubility also affect the dynamics of the compounds (*Szöcs et al., 2017*).

The ratio of parent compounds and TPs were found to be different between pesticides and creeks. These fluctuations of the ratio of parent compound to their respective TP were also highlighted by *Halbach et al. (2021)*, in which the concentration of the terbuthylazine metabolite exceeded the value of its parent compound during a rainfall event, whereas the concentrations of the metabolites of (s)-metolachlor were not elevated. The monitored fluctuation of concentrations of the parent compound and its respective TP in small agricultural creeks highlight the importance of taking characteristics such as the distance of fields to the water body, the morphology of the catchment size or the presence of drainage systems into account to provide information about the transmission pathway (*Halbach et al., 2021*; *Gassmann et al., 2013*).

We ascertained that high peak concentrations of pesticides mostly occur within the first two hours after the start of sampling period, *i.e.,* during the first hours of water level increase due to the respective rain event. This underlines that rainfall events can lead to high peak concentrations of pesticides in small creeks, as mentioned in other previous studies (*Halbach et al., 2021*; *Bundschuh, Goedkoop & Kreuger, 2014*; *Schulz, 2004*). *Spycher et al. (2018)* confirmed a strong temporal coincidence of rain events and concentration peaks and concluded that rain-triggered runoff transport is an important pathway for pesticides. However, the concentration of the TPs like metazachlor-ESA and metazachlor-OA mainly increased within a sampling cycle leading to the assumption that they also reach surface waters by exfiltration of surface-near groundwater (*Kern et al., 2011*) (Fig. S3).

## Frequency and duration of RAC exceedances

The RAC value is defined as the environmental concentration above which unacceptable effects on the environment cannot be excluded (*EFSA Panel on Plant Protection Products and their Residues (PPR), 2013*). In the course of the approval of plant protection products (PPPs), care is taken to prevent RAC exceedances by prescribing farmers how to apply the PPPs in the fields. By respecting certain risk management measures—such as keeping distance to water bodies during the application or increasing buffer strips—the risk of RAC exceedances shall be avoided (*Liess et al., 2021*; *Bereswill, Streloke & Schulz, 2013*). Nevertheless, studies have shown that especially small creeks are exposed to ecologically relevant pesticide concentrations(*Szöcs et al., 2017*), which may affect invertebrate communities (*Liess & Von Der Ohe, 2005*). *Liess et al. (2021)* found that agriculturally derived pesticides are the main cause of declines in sensitive insects in aquatic invertebrate communities, even at concentrations below current thresholds. *Stehle & Schulz (2015a)* also determined that insecticide levels higher than legally accepted regulatory threshold levels pose a high risk to freshwater biodiversity. Smaller-scale studies confirmed the observations of pesticide-induced adverse effects on ecosystems in small surface waters in correlation with agricultural land use (*Bereswill, Streloke & Schulz, 2013*; *Schäfer et al., 2012*).

In our study, RAC exceedances occurred in 55% of rainfall events at all three sampling sites. Exceedances were mainly detected for the insecticide thiacloprid (up to 176-fold) followed by the herbicides diflufenican, (S)-metolachlor, prosulfocarb and terbuthylazine. The highest number and longest period of RAC exceedances per sampling cycle occurred in May, 2017 and May, 2018 (Table S9). This is consistent with the main application period of PPPs in spring (*Kreuger, 1998*). *Szöcs et al. (2017)* and *Halbach et al. (2021)* observed similar results, whereby neonicotinoid insecticides—especially thiacloprid—showed also high RAC exceedances. However, it should be noted that concentrations of the insecticide thiacloprid in the environment are expected to decrease in the future as the authorization of thiacloprid was not renewed by the EU on January 13, 2020 and its use is banned from February 3, 2020 (*European Union, 2020*).

By contrast, a pesticide monitoring in Switzerland during the years 2005 to 2012 found RAC exceedances mainly for herbicides and fungicides in surface waters, while only one insecticide was above the RAC (*Knauer, 2016*). Moreover, RAC exceedances occur more often in small creeks flowing through intensively used arable land compared to rivers (*Stehle & Schulz, 2015b*). Additionally, *Halbach et al. (2021)* showed that RAC exceedances not only occur during rain events, but also during dry weather periods. Nonetheless, during rain events RAC exceedances rise from 23% (at 50% of the analyzed sites) to 60% of the samples(at 73% of the analyzed sites) (*Halbach et al., 2021*). We observed that RAC exceedances (*e.g.*, of (S)-metolachlor, thiacloprid, terbuthylazine) lasted up to 24 h after heavy rain events. Accordingly, RAC exceedances occur regularly for extended time periods, although their frequency and duration depend on the weather conditions.

Thus, the current risk assessment schemes and their accompanied risk mitigation measures are not sufficient to avoid RAC exceedances and consequently protect the aquatic ecosystem. Moreover, it should not be overlooked that pesticide mixtures enter the creek.

Therefore, the protectiveness of the RAC value must be discussed, because it is defined for a single active compound only (*EFSA Panel on Plant Protection Products and their Residues (PPR), 2013*). Combined additive or even synergetic effects of pesticide mixtures are not included but should be taken into account (*Denton et al., 2003*; *Belden & Lydy, 2006*; *Stehle & Schulz, 2015b*).

## Baseline toxicity and toxic units

The Microtox assay is a common method to assess the toxicity of the water samples (*Tang et al., 2013*). Bacterial bioluminescence is proportional to metabolic activity, resulting in a decrease in case of a disruption by toxic substances (*ISO 11348-3, 2007*). As considered, the $EC_{50}$ is the effect concentration, which causes 50% bioluminescence inhibition of *A. fischeri* referring to the REF. Consequently, the smaller the value of $EC_{50}$ REF, the higher the toxicity. In this study, several enriched extracts caused a significant baseline toxicity. *De Zwart & Slooff (1983)* compared the sensitivity of *Aliivibrio fischeri* with other standard aquatic toxicity tests for 15 organic and inorganic chemicals and came to comparable results. This suggests that the baseline toxicity found in this study poses a risk to the aquatic community in the monitored creeks. The highest baseline toxicity was measured during a heavy rain event in the water samples collected at the time of the highest water level and thus with the highest concentrations of pesticides and then steadily decreased over time as the water level decreased. It can therefore be assumed that during rain events that occur immediately after a pesticide application, even higher amounts of pesticides than determined in this study enter the creek and consequently lead to high peak concentrations (*Spycher et al., 2018*). Our study shows that the duration of RAC exceedances after a heavy rain event, which are likely to result in adverse effects on the aquatic biocenosis, lasted up to 24 h. *Raby et al. (2018)* investigated the potential latent effects of a single pulse (duration of 24 h) of the insecticides imidacloprid and thiamethoxam on early life stages of four aquatic arthropods. At the highest imidacloprid concentrations (8.8 and 8.9 µg/L) immobilization was found after a single 24 h exposure in two of the four species. However, a recovery was observed during the post treatment-period so that no long-term effects remained (*Raby et al., 2018*). With the increasing frequency of rain events under the conditions of climate change, it can be assumed that the recovery potential of organisms will be weakened more in the future. Furthermore, organisms will have to cope with a variety of additional stressors, such as thermal irregularities or drought events due to climate change. *Verheyen & Stoks (2023)* tested the effects of the pesticide chlorpyrifos on low- and high-altitude populations of *Ischnura elegans* damselfly larvae across a range of temperatures. It was found that chlorpyrifos was not toxic to damselfly larvae at mean temperatures but became more toxic at lower and higher average temperatures. Therefore, it can be expected that not only short-term but also long-term effects on aquatic environment will be more likely.

*Boxall et al. (2013)* compared the effects of pulsed (two- and four-days scenario) and continuous exposures on the growth of aquatic macrophytes. At the highest tested concentration of metsulfuron-methyl, the exposure of pulses showed similar effects compared to the continuous exposure. Furthermore, the negative effect of pentachlorophenol on the growth of macrophytes was even greater for the pulsed compared

to the continuous exposure (*Boxall et al., 2013*). Consequently, the repeated occurrence of high peak pesticide concentrations in small creeks is likely to result in a negative impact on organisms and the environment, depending on their frequency and duration.

By comparing the results of the Microtox assay and the calculated STUs it is conspicuous that the Langder Flutgraben samples were the most toxic. Waschbach and Weidgraben samples were less toxic.

To verify the hypothesis that the observed biological response correlates with the STUs of each creek, a Pearson correlation was conducted. By comparing the $EC_{50}$ REF with the STUs calculated for each creek, the strongest correlation was found for Langder Flutgraben (Fig. 7). This significant correlation further supports that pesticide inputs in small creeks can cause considerably baseline toxicity and have the potential to affect the creek ecosystem. Due to their strong response to pesticide contaminations, macro-invertebrates are applied as indicator organisms in the SPEAR$_{index}$ (*Liess & Von Der Ohe, 2005*). By using the SPEAR$_{index}$, changes of community structures were observed until a TU level down to −3 (*Schäfer, Brink & Liess, 2011*). Additionally, *Liess & Von Der Ohe (2005)* observed that the abundance of SPEAR was reduced by 60% from April until May at sampling sites where TU values were between −1 and −2. According to the STUs and RAC exceedances measured in this study, corresponding effects on the algae/aquatic plants and invertebrate community are to be expected. Thus, effects on non-target organisms in small creeks based on pesticide inputs from agricultural land are widely distributed. This may lead to a shift within a benthic community resulting in a decline of species at risk while robust species become more widespread.

## CONCLUSIONS

These results indicate that current pesticide monitoring programs are likely to underestimate pesticide risks to water bodies and their aquatic biocenosis because they typically do not consider event-based sampling. In particular, the pesticide input to small agricultural creeks is not well reflected in official monitoring programs. Current pesticide monitoring programs count on grab sampling taken monthly or weekly, while precipitation events, the landscape and morphological characteristics are not considered (*Szöcs et al., 2017*; *Leu et al., 2004*). For the future, it is suggested that small water bodies should be included into official monitoring programs which, as far as possible, take the high event-driven variability of pesticide emissions into account, especially during the main application periods. These recommendations do not only apply to Germany, but also to regions with comparable climate conditions and cultivation practices. Also, our results show that rainfall events can lead to a considerable increase of pesticide concentrations, partly exceeding RACs and baseline toxicity. This indicates that high peak concentrations can lead to toxic effects in the aquatic biocenosis. Since there are studies assessing the correlation between pulse peak concentrations and toxic effects to different aquatic organisms (*Raby et al., 2018*; *Boxall et al., 2013*), it is necessary for the future to investigate this correlation adding more natural stressors by using multi-stressor test systems to create more realistic conditions. Moreover, it is important to evaluate an ecotoxicological assessment to also

adequately monitor effects of the aquatic community as well because our observations have shown that the current risk mitigation measures are insufficient to protect the aquatic environment.

## ACKNOWLEDGEMENTS

The authors thank Andrea Dombrowski (Goethe University Frankfurt) and Liza-Marie Beckers (BfG) for their technical and helpful support.

### Funding
The authors received no funding for this work.

### Competing Interests
Jörg Oehlmann is an Academic Editor for PeerJ. Arne Wick and Björn Jacob are employees of the German Federal Institute of Hydrology, Koblenz, Germany.

### Author Contributions
- Sarah Betz-Koch analyzed the data, prepared figures and/or tables, authored or reviewed drafts of the article, and approved the final draft.
- Björn Jacobs performed the experiments, authored or reviewed drafts of the article, and approved the final draft.
- Jörg Oehlmann conceived and designed the experiments, authored or reviewed drafts of the article, and approved the final draft.
- Dominik Ratz performed the experiments, authored or reviewed drafts of the article, and approved the final draft.
- Christian Reutter performed the experiments, authored or reviewed drafts of the article, and approved the final draft.
- Arne Wick conceived and designed the experiments, prepared figures and/or tables, authored or reviewed drafts of the article, and approved the final draft.
- Matthias Oetken conceived and designed the experiments, authored or reviewed drafts of the article, and approved the final draft.

### Data Availability
 The raw measurements are available in the Supplementary File.

### Supplemental Information
Supplemental information for this article can be found online at http://dx.doi.org/10.7717/peerj.15650#supplemental-information.

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
