# Peer review of "Pesticide dynamics in three small agricultural creeks in Hesse, Germany"

_PeerJ, doi:10.7717/peerj.15650_

## Round 0.1 · original submission · Major Revisions

Both reviewers have made numerous suggestions to improve your manuscript, notably, both have indicated a need to acknowledge extensive previous work in this field (pesticide impacts on small catchments / agricultural catchments), as some elements of novelty in your study are contested. I would expect all of these suggestions to be addressed in your revision and rebuttal.

Reviewer 1 ·

Basic reporting

Context:
I would appreciate if the results were put into a bigger frame by addressing the following questions in the manuscript: Are these results transferable to other regions? Since high herbicide concentrations were measured: What are the expected effects on other organism groups besides invertebrates/What might TUs for fish and algae look like? Is the analyte spectrum expected to cover the majority of pesticide toxicity with regard to other pesticides used in agriculture nowadays? Are the rain events monitored considered extreme events or typical rain events? Based on other literature: Are there other peak exposure events besides the ones induced by rain and how relevant are these (e.g. spray drift)? What is the meaning of the derived REFs from an ecological point of view?

Experimental design

no comment

Validity of the findings

no comment

Additional comments

Line 19 & 22: The authors state that there’s a knowledge gap since streams with catchments < 10 km2 are not regularly monitored. However, the streams you monitored are also < 10 km2. This is no critique of your approach, but maybe rephrase so that it does not sound so contradictory. The actual problem ist that even streams of the size you investigated are only rarely part of the EU-WFD.
Line 55: unsatisfactory chemical status? For the ecological status, other factors like river morphology are expected to play a major role.
Line 81: Better provide full name for RAC when mentioning it for the first time in the main text.
Line 82: It’s not “through the provision of RACs”, but rather through the concept of only authorizing uses that are predicted to result in surface water concentrations not exceeding the RAC, that unacceptable effects should be avoided.
Line 203: The authors also measured pesticides, where Daphnia magna clearly is not the most sensitive species, particularly neonicotinoids. Since the authors aim “to compare the toxicity of the measured pesticide concentrations”, using chironomid (=insect) toxicity data where respective effect concentrations are lower than for Daphnia magna is necessary.
Line 211: Only after heavy rainfalls?
Line 255: Tukey, not Turkey
Line 268: pesticides detected
Line 316: Sentence not fully clear: “Overall, RAC exceedances of maximum concentrations were detected in 11 out of 20 sampling events corresponding 55 % at three sampling sites”. Do the authors mean “Overall, RAC exceedances were detected at all three sampling sites in 55 % of sampling events (11 out of 20)”?
Line 318: Provide information in the text for how many substances measured RACs were available in line 236f.
Line 335: Remove “by”
Line 348: Please indicate number of CS included in each box in Figure 5.
Line 371: Maybe the correlation increases when including chironomid tox data.
Line 374: The following chapter covers more than just concentration ranges (seasonal/spatial), which is why I suggest to broaden its header. E.g. exposure patterns of pesticides
Line 376: It’s worth to mentioned that Halbach et al. also performed event-driven samplings.
Line 379: flufenacet
Line 400: were instead of was
Line 417: It is worth to mention that, in general, measured concentration levels seem to follow the respective application patterns rather well. Insecticides are almost exclusively applied in spring/early summer, while herbicides and fungicides are also widely applied in winter cereals in autumn. Similar observations were recently published by Vormeier et al. 2023: Temporal scales of pesticide exposure and risks in German small streams, which should be mentioned due to the overlap of research.
Line 419: detection frequency
Line 438: Decide whether it’s written run-off (further above) or runoff
Line 442: et al.?
Line 445: that is not correct, as short-term effects can be classified as acceptable. More correct: above which unacceptable effects on the environment cannot be excluded.
Line 449: PPPs must not be applied during rain at any time, that is no specific risk management measure
Line 453: The cited paper of Stehle and Schulz is not best choice for this statement: I propose Stehle and Schulz 2015: Agricultural insecticides threaten surface waters at the global scale (Same authors, same year) and the already elsewhere cited paper of Liess et al. 2021, where the actual ecological status is linked to the factor of RAC exceedance. Liess et al. even show that concentrations below the RAC cause effects classifiable as unacceptable.
Line 458: worth a mention: at all three
Line 506: Especially also due to rising water temperatures and drought events causing additional stress
Line 532: worth a mention: …and according to the STUs and insecticide RAC exceedances measured, respective effects are to be expected on the invertebrate community.
Line 539: “although” not suitable here. “while”?
Line 547: There are many more that assess “the correlation between pulse peak concentrations and toxic effects to different aquatic organisms”, please be more specific then.
Line 549: What is meant by “note”? “Adequately monitor” maybe? Please be more specific
Figure 6: Please include number of assays per box
Table 1: Second column decimals given with commas. Also, why is there no catchment area given for the Weidgraben?
Regarding analyte selection: Please provide statement regarding to what extent pesticide toxicity may have been covered. For example glyphosate, the most prominent herbicide, is not included, but may significantly increase the in-stream toxicity.

·

Basic reporting

The manuscript titled « Pesticide dynamics in three small agricultural creeks in Hesse, Germany» provides reliable monitoring data contributing to the goal of protecting aquatic environments. The manuscript is very well written, it follows a clear/logical structure and uses solid references. Furthermore, raw data are supplied so that they are FAIR: Findable, Accessible, Interoperable and Reproducible.

Suggestions for improvement:
L53-54 «water bodies exceed the quality standard» – it reads like these water bodies are doing well. Is it what the Authors mean? Or perhaps would be better to rephrase in case the Authors mean that safety regulatory values are exceeded thus leading to a degraded environment?
L65 The Authors state: «In particular, the dynamics of pesticide exposure, especially the concentration courses after or during rain events and the seasonal variations, in small creeks is largely uninvestigated, […]». But actually, recent innovative works are filling these knowledge gaps, such as in la Cecilia et al (2021) (reviewer paper on agricultural catchment in wet periods), la Cecilia et al (2023) (reviewer paper on agricultural catchment in dry periods) and Stravs et al. (2021) (relevant also for urban-impacted catchments).
la Cecilia, D., Dax, A., Ehmann, H., Koster, M., Singer, H., and Stamm, C. (2021). Continuous high-frequency pesticide monitoring to observe the unexpected and the overlooked. Water Res. X. 13, 100125. doi: 10.1016/j.wroa.2021.100125 (reviewer paper)
la Cecilia D, Dax A, Ehmann H, Koster M, Singer H and Stamm C (2023) Continuous high-frequency pesticide monitoring in a small tile-drained agricultural stream to reveal diel concentration fluctuations in dry periods. Front. Water 4:1062198. doi: 10.3389/frwa.2022.1062198 (reviewer paper)
Stravs, M.A., Stamm, C., Ort, C., Singer, H., 2021. Transportable Automated HRMS Platform “MS2field” Enables Insights into Water-Quality Dynamics in Real Time. Environmental Science & Technology Letters 8 (5), 373–380. https://doi.org/10.1021/acs.estlett.1c00066.
L93 From this line, the Authors briefly introduce the topic of toxicity in aquatic environments. The approach of Toxic Units is presented at L106 but no reference is made to the second approach called «microtox». Perhaps this aspect should be solved.
L107 The research hypothesis “Rainfall events lead to high peak concentrations of pesticide in small creeks due to their vicinity to agricultural land” is not investigated here because “vicinity” is not a variable of the problem. I would suggest to rephrase on the following line “rain events activate contamination sources and drive contamination of nearby streams”. Moreover, the Authors will reveal only later in the manuscript that 2 out of 3 streams are impacted by urban activities. And yet, the agricultural stream will have lower concentrations that the urban-affected ones. Therefore, the Authors might want to include the focus on contamination from the urban landscape and not just from agriculture. This is an interesting aspect considering that cities are growing and expanding.

In the current state, figures are of poor quality. I cannot read labels and if I zoom in, the figure becomes pixelated.
I miss to see a figure showing the boundary conditions (i.e. rainfall and water level), if they were continuous. This would give an indication on the dynamicity of the catchments.
Figure 2 can be more informative, like showing boxplot of concentrations per catchment and per pesticide group. Tables 4 and 5 are not necessary, if Figure 2 is improved.
Figure 3 again, more informative if divided by catchment.
Figure 4 Please explain the meaning of the letter in the caption (A B C D…). Do they refer to the panels? Then, it would be: Panel A, …
What are the concentration units? ng/l? I wuold suggest to be consistent with the units presented in the manuscript, i.e. microg/l.
Why different colors for dots and lines (I assume for compounds and RAC? – cannot read).
Figure 8 is linear the best curve? Why not a dose-response non-linear regression?
Table 1 there is no catchment area for the third catchment.

Experimental design

The chosen approaches for water monitoring and laboratory analysis are reliable. The Authors formulated a clear hypothesis and the data collected, as well as the data analysis, are mature to generate results needed to assess the validity of the hypothesis.

Suggestions for improvement:
In the Introduction Section, The Authors stress that the European Union Water Framework Directive (WFD) does not include small catchments (area < 10 km2) in mandatory monitoring. However, in the Methods Section, it is explained that the studied catchments are (still small but) double the size (i.e., 25.8 km2, 26.4 km2, and actually missing information for the third one – Table 1). Could please the Authors explain. Would the studied catchments with their size be considered in the WFD? If yes, why the focus in the Introduction Section on small catchments?
L140 there are CSO. This should be mentioned in the Introduction Section and the consequent impacts briefly mentioned.
L144 Please specify if the sampler is not refrigerated. If so, could there be any degradation in the summer months over the 12 hours (L160) needed to collect the samples?
L153 The above suggested reference (la Cecilia et al 2020 – reviewer paper) clearly shows the dynamics of many Plant Protection Products in an agricultural catchment continuously at 20 minutes resolution.
L160 not clear, why having a detailed sampling scheme if then 3 bottles were mixed? Were they kept separated for chemical analyses and then mixed for the toxicity test?
L181 The sampling was done in 2017 and 2018, but when were the chemical analysis performed?
L188 How many and which compounds were targeted (this information could go in the SI)?
L198 Were Isotopes and Standards used to compensate for matrix effects?
L202 why Daphnia and not other non-organisms?
L212 are concentrations really additive in their effects? Or maybe it would be better to state such assumption?
L238 “RAC due to”, maybe better to rephrase as “RAC based on”?
L239 not clear how the mean was calculated, it would be clear if the equation used is written. I would suggest to give a specific name to this variable (like C with a hat = sum(Ct * t)/sum(t), if I understood well). All samples collected were considered in the mean?

Validity of the findings

L275 Is this the new variable “C with a hat”?
L287 Would you need here to define “C with a hat and subscript winter” to indicate the winter period, and same for summer?
L289 Why these two catchments were used and not three? I see in Table 2 that for “Langder Flugraben” there are no samples in 2017? Is it because there were no rainfall events in 2017 as stated in Table 3? Is it possible that there were no rainfalls in one catchment but yes in the nearby catchments?
L310 why figure 4 is presented before figure 3 (at L319)?
L315 It is true that the dynamics of transformation products (TPs) can be different from those of their parent compounds. This was clearly observed in the above mentioned paper from the reviewer (la Cecilia et al 2020).
L317 corresponding “to” 55% ? If yes, please add. Same as L321 to L325.
L334 decreasing “in” concentrations? If yes, please add.
L370 correlation is not shown in Figure 7. Maybe the Authors meant Figure 8 which is not mentioned in the manuscript?
L385 Was it a coincidence that concentration values in Kreuger (1998) match the here reported numbers? Or can this be attributed to similar land use and land management?
L403 Odd sentence. “due to Müller at al. (2002)” or “due to processes as … as shown by Müller at al. (2002)”?
L405 It is okay not to quantify the contaminants load from the CSO in this work if the focus is on all the land uses that can contribute to water contamination. It would be essential if the focus of the research is to assess the role of agriculture in water contamination. This is why it is important to develop a research hypothesis that can be investigated with the available data.
L411 “at the creek” or “in the creek”?
L546 I would recommend the following articles to more robustly conclude on the effect of pulsed concentrations (Ashauer et al 2013 and Ashauer et al 2016) as well as to assess the role of abiotic factors (like temperature) or the toxicity of pesticides (Verheyen & Stoks, 2023).
Ashauer, R., Brown, C.D., 2013. Highly time-variable exposure to chemicals–toward an assessment strategy. Integr Environ Assess Manag 9 (3), e27–e33. https://doi.org/10.1002/ieam.1421.
Ashauer, R., Brown, C.D., 2013. Highly time-variable exposure to chemicals–toward an assessment strategy. Integr Environ Assess Manag 9 (3), e27–e33. https://doi.org/10.1002/ieam.1421.
Ashauer, R., Albert, C., Augustine, S., Cedergreen, N., Charles, S., Ducrot, V., Preuss, T. G., 2016. Modelling survival: exposure pattern, species sensitivity and uncertainty. Sci. Rep. 6, 29178. https://doi.org/10.1038/srep29178.
Verheyen, J. & Stoks, R, (2023). Thermal Performance Curves in a Polluted World: Too Cold and Too Hot Temperatures Synergistically Increase Pesticide Toxicity. Environ. Sci. Technol., 57, 3270-3279, 10.1021/acs.est.2c07567
Finally, just a remark, pesticides can be used in forestry, which I see occupy quite a portion of the Weidgraben catchment.

Additional comments

I feel like the suggested revisions would improve the clarity and the visualization of the data collected.

---

## Round 0.2 · accepted · Accept

Both reviewers have indicated that all issues raised during the review process have been addressed, and your manuscript is now ready for publication. They also note a couple of minor typos which can be corrected at the proof stage.

Reviewer 1 ·

Basic reporting

no comment

Experimental design

no comment

Validity of the findings

Optional: I suggest providing the collected EC50 values from the PPDB for TU calculation in form of a supplementary table. This enables a validation of results even if the values in the PPDB will be updated in the future.

Additional comments

no comment

·

Basic reporting

no comment

Experimental design

no comment

Validity of the findings

no comment

Additional comments

I previously reviewed the revised manuscript. The authors implemented all the feedbacks from the two reviewers and transparently stated the limitations of their work.
Just two typos. In the track change document, at L67 "Framwork" is "Framework" and L72 and 729 "La Cecilia" is "la Cecilia".
Keep up the good work.